# Thermomechanical and Alkaline Peroxide Mechanical Pulping of Lignocellulose Residue Obtained from the 2-Furaldehyde Production Process

**DOI:** 10.3390/ma15175872

**Published:** 2022-08-25

**Authors:** Maris Puke, Daniela Godina, Prans Brazdausks, Janis Rizikovs, Velta Fridrihsone

**Affiliations:** 1Latvian State Institute of Wood Chemistry, Dzerbenes 27, LV-1006 Riga, Latvia; 2Department of Chemistry, Latvia University of Latvia, Jelgavas 1, LV-1004 Riga, Latvia

**Keywords:** birch wood, lignocellulose, thermomechanical pulping process, alkaline peroxide mechanical pulping process

## Abstract

The necessity for the reduction in greenhouse gas emissions, the growing demand for the improvement of biorefinery technologies, and the development of new biorefining concepts oblige us as a society, and particularly us, as scientists, to develop novel biorefinery approaches. The purpose of this study is to thoroughly evaluate the leftover lignocellulosic (LC) biomass obtained after the manufacture of 2-furaldehyde, with the intention of further valorizing this resource. This study demonstrates that by using thermomechanical and alkaline peroxide mechanical pulping techniques, birch wood chips can be used in the new biorefinery processing chain for the production of 2-furaraldehyde, acetic acid, and cellulose pulp. In addition, the obtained lignocellulosic residue is also characterized. To produce a lignocellulosic material without pentoses and with the greatest amount of cellulose fiber preserved for future use, a novel bench-scale reactor technology is used. Studies were conducted utilizing orthophosphoric acid as a catalyst to deacetylate and dehydrate pentose monosaccharides found in birch wood, converting them to 2-furaldehyde and acetic acid. The results showed that, with the least amount of admixtures, the yields of the initial feedstock’s oven-dried mass (o.d.m.) of 2-furaldehyde, acetic acid, and lignocellulose residue ranged from 0.04 to 10.84%, 0.51 to 6.50%, and 68.13 to 98.07%, respectively, depending on the pretreatment conditions utilized. The ideal 2-furaldehyde production conditions with reference to the purity and usability of cellulose in residual lignocellulosic material were also discovered through experimental testing. The experiment that produced the best results in terms of 2-furaldehyde yield and purity of residual lignocellulose used a catalyst concentration of 70%, a catalyst quantity of 4%, a reaction temperature of 175 °C, and a treatment period of 60 min. It was possible to create pulp with a tensile index similar to standard printing paper by mechanically pulping the necessary LC residue with alkaline peroxide, proving that stepwise 2-furaldehyde production may be carried out with subsequent pulping to provide a variety of value-added goods.

## 1. Introduction

European incentives to speed up the shift from a petro-based economy to a robust and uniform biobased economy have dramatically increased during the preceding ten years [1]. With the European Carbon Law, the EU has established a legally binding goal for the EU to become climate-neutral by 2050, as part of the European Green Deal. In order to achieve this, greenhouse gas emissions must significantly decline during the following decades. The EU has increased its 2030 climate ambition, aiming to reduce emissions by at least 55% by 2030, as a first step toward achieving climate neutrality. As part of the so-called “Fit for 55 package,” the EU is striving to update its regulations relating to the environment, energy, and transportation in order to bring them into line with its 2030 and 2050 goals. The new legislation includes a number of proposals for achieving this goal. The legislative ideas and policy actions included in the Fit for 55 package are as follows: (1) the EU’s emissions trading system, (2) member states’ emission reduction targets, (3) emissions and removals from land use, land use change, and forestry, (4) renewable energy, (5) energy efficiency, (6) alternative fuels infrastructure, (7) CO_2_ emission standards for cars and vans, (8) energy taxation, (9) the carbon border, (10) sustainable aviation fuels, (11) greener fuels in shipping, and (12) the social climate fund [2]. To achieve these goals, Europe must move away from fossil resources and more readily adopt the widespread use of renewable resources. 

The biomass-based feedstock that is most widely available is LC biomass, a renewable resource [3] that does not compete with the food supply. Furthermore, significant amounts of LC biomass are frequently produced from agricultural and forestry wastes. For this reason, LC biomass is essential as a source of biofuels and other high-value compounds, such 2-furaldehyde, lactic acid, and monomeric phenols, which can be employed as building blocks in the production of polymeric materials [4,5]. Additionally, LC biomass can be hydrolyzed using an enzymatic or acid catalyzed method to yield chemical resources such as ethanol, reducing sugars, and 2-furaldehyde. Theoretically, any material containing pentoses might be used as a raw material for the manufacture of 2-furaldehyde; however, for commercially viable production, a biomass must include a minimum of 15 to 20% pentoses. With the current production methods, only about one-third of the pentosanes in raw materials can be transformed into 2-furaldehyde. The high demand for its derivatives, particularly 2-furaldehyde alcohol, which is used primarily in the production of furan resins for foundry sand binders and is thought to be the main market for 2-furaldehyde, makes 2-furaldehyde an important chemical because it is a selective solvent for separating saturated and unsaturated compounds in petroleum, gas, oil, and diesel fuel refining [6]. Furthermore, 2-furaldehyde can be used as a starting material to make a variety of compounds, including methylfuran, furfuryl alcohol, tetrahydrofurfuryl alcohol, tetrahydro-furan, methyltetrahydrofuran, dihydropyran, and furoic acid. The most versatile reaction for upgrading furanic compounds is hydrogenation of the aldehyde group or furan ring, which can be used to make hydrocarbon fuels directly from furan derivatives. Hydrogenolysis can create alcohols such as 1,5-pentanediol by cleaving the furan ring. Adduct creation via aldol condensation and dimerization, followed by hydrodeoxygenation, can yield C8 to C13, as well as longer chain hydrocarbons, from 2-furaldehyde [7,8,9]. One of the top 30 biobased compounds has been identified as 2-furaldehyde [10].

As a result, it is critical to integrate the manufacture of 2-furaldehyde and bioethanol or other products into a single process, but this has been discussed only theoretically so far. A large percentage (40–50%) of cellulose is destroyed during the pretreatment process in all known 2-furaldehyde manufacturing processes [11]. The pretreatment processes attempt to optimize the separation of the lignocellulosic biomass (cellulose, hemicellulose, and lignin) to obtain bleached-cellulose and thus increase the yield of second-generation bioethanol.

Because of Latvia’s climatic circumstances, birch wood (*Betula pendula Roth*.) is the most commonly utilized tree species for plywood manufacture. Birch stands cover 27% of Latvian forestland, or 881 thousand hectares, and they are a good source of cellulose and hemicellulose based raw materials [12]. One of the leading plywood producers in Eastern Europe, Latvia produces 250,000 m^3^ of plywood annually. Up to 30% of the processed wood is made up of low-value byproducts, such veneer shorts, cores, and cut-offs [13], all of which offer a high potential for the development of value-added products. As a result, we selected this material as a viable source of 2-furaldehyde, acetic acid, and fibers. The Latvian State Institute of Wood Chemistry Biorefinery Laboratory has an original pilot plant in which it is possible to study the industrial 2-furaldehyde-acetic acid-obtaining process. As for the methods for obtaining thermomechanical pulp (TMP) and alkali peroxide mechanical pulp (APMP), these are widely described in the literature, but the use of these methods for obtaining 2-furaldehyde from the residue of LC has not yet been studied. The advantages of the APMP process are that the wood chips can be fully bleached prior to refining, there is a high flexibility in processing different types of wood, and the process delivers fibers with higher density, tear, and tensile strength compared to TMP fibers [14]. 

Because of this, the purpose of this study is to offer information on the acquired 2-furaldehyde, acetic acid, and LC residue that is left over after 2-furaldehyde is produced, as well as information on its composition and potential for use as a feedstock in TMP and APMP. The study’s objective is to obtain 2-furaldehyde at levels greater than 60% of the theoretical maximum with less than 10% cellulose degradation. A theoretical foundation for the creation of industrial technology, as well as experimental methods that will show the feasibility of implementing industrial technology, will be researched. The approach will involve the preparation of deciduous wood, which will yield useful byproducts such as 2-furaldehyde, acetic acid, and cellulose fibers. Final products that can be produced commercially include 2-furaldehyde and acetic acid. Obtained LC residue will be investigated as a feedstock for TMP and APMP processes, to obtain pulps with properties comparable to commercial products. 

## 2. Materials and Methods

Figure 1 is a schematic diagram illustrating the bigger picture of the study, emphasizing the experimental work completed. It also provides a look back at our previous publications, which were a starting point of our study.

### 2.1. Materials and Chemicals

All of the used chemicals were purchased from Merck (Germany) and used without further purification. These chemicals included orthophosphoric acid (H_3_PO_4_) (85%), sulfuric acid (H_2_SO_4_) (95–97%), barium carbonate (BaCO_3_) (99%), sodium hydroxide (NaOH) (98%), hydrogen peroxide (H_2_O_2_) (30%), sodium thiosulfate (Na_2_SiO_3_) (98%), D-(+)-cellobiose (≥99%), D-(+)-glucose, (≥95%), D-(+)-xylose (≥99%), L-(+)-arabinose (≥99%), D-(+)-galactose (≥99%), D-(+)-mannose (≥99%), 2-furaldehyde (≥99%), acetic acid (≥99%), 5-hydroxymethylfurfural (5-HMF) (≥99%), levulinic acid (≥98%), and formic acid (≥95%).

### 2.2. Samples

The A/S Latvijas Finieris company “Lignums,” which specializes in the creation of plywood and processed wood chips, provided the birch wood chips (BWCs). The business provides BWCs to Scandinavian pulp manufacturers. We are utilizing common wood chips, which are created in cellulose in pulp mills. In Figure 2, the fractional distribution of used birch chips is displayed.

BWCs were procured, air-dried, and stored at 15–20 °C to avoid degradation before being processed further. In the lab, the relative humidity ranged from 25 to 35%. BWCs had particles that ranged in size from 45 to 47 mm.

### 2.3. Catalyzed Pre-Treatment of BWC

A scheme of the processes of catalyzed pre-treatment of BWC and the pulping process, including the parameters and their values, is shown in Figure 3.

In a blade-type mixer of unique design, BWCs (particle size 45–47 mm and moisture content Wrel = 40.43%) were mixed in a catalyst solution (H_3_PO_4_). An H_3_PO_4_ acid solution of a varied concentration (55–85%) was used as a catalyst. After combining the chips with a certain amount of the catalyst (3 wt.%), the resulting material was treated with a continuous superheated steam flow in an original bench-scale reactor, as detailed in our earlier papers [15,16]. The diameter of the main reactor camera was 110 mm, its height was 1450 mm, its volume was 13.7 L, and its max pressure was 1.2 MPa [17]. To provide a constant temperature in the reaction zone throughout the entire process period and with various process parameters, the reactor included two automatic heat insulation systems, as indicated in Table 1. A total of 8 experiments were performed. The steam leaving the reactor was condensed, and samples were taken every 10 min. The condensate contained mainly a water solution of 2-furaldehyde, acetic acid, levulinic acid, 5-HMF, and formic acid. The LC from the steam-treated wood chips, containing mostly lignin and carbohydrates, was discharged from the reactor. Wet chemistry analytical procedures specified in the Technical Association of the Pulp and Paper Industry (TAPPI) standards were used to determine the chemical composition of the BWC [18,19,20]. Calculations for product yields and catalyst quantities were based on the initial feedstock’s oven-dried mass (o.d.m.). Three parallel tests were run for each sample, and the average of the data is displayed with a relative standard deviation (RSD) of less than 5% for all experiments.

### 2.4. HPLC Analysis

Using a Shimadzu LC-20A HPLC (Shimadzu, Tokyo, Japan) with a refractive index detector, the amount of monosaccharides, 2-furaldehyde, 5-HMF, and organic acids in the resulting hydrolysates were quantified. Cellobiose, glucose, xylose, arabinose, galactose, mannose, 2-furaldehyde, acetic acid, 5-HMF, levulinic acid, and formic acid were used as reference standards. We employed a Shodex Sugar SH1821 column at 60 °C, with an eluent of 0.008 M H_2_SO_4_ at a rate of 0.6 mL/min^-1^, to separate the cellobiose, glucose, 2-furaldehyde, acetic acid, 5-HMF, levulinic acid, and formic acid. For the carbohydrate analysis, we used a Shodex Sugar SP0810 column at 80 °C, with deionized water as the mobile phase under a flow rate of 0.6 mL·min^−1^. Samples were neutralized to pH 5–7 with BaCO_3_ and filtered through a 0.2 μm membrane filter before injection. All samples were tested three times. For each analyzed standard, the equations for the calibration curves are given in our previous publication [15].

### 2.5. Pulping Process of LC Residue Obtained after BWC Pre-Treatment

The LC residue left over after obtaining 2-furaldehyde was dried to a moisture content of 4–8%. In the case of TMP, the MD-3000 disc refiner was filled with water. The liquid to wood ratio (L/W) was 4:1, and the total residue of the LC chips was fiberized, gradually reducing the gap between the refining discs until the size reached 2.5 mm. In the case of APMP, the best variant of the APMP process was found, based on the literature review [14]. Prepared chemical charges (%/o.d.m.) that were used during the two-stage impregnation (Table 2). In the first stage, the necessary chemicals were added to the LC residue on the hob in the pot and kept at 80–90 °C for 60 min, with periodic stirring. In the second stage, the necessary chemicals (Table 2) were added during the fiberization process. In all experiments, the refining time was 5 min. The resulting fibers were then drained from the excess solution, rinsed, and dried for the casting of paper discs; the fiber strength was then analyzed.

Standard handsheets were made according to ISO 5269/2 standards by a PTA “Rapid-Köthen” handsheets paper machine (PTI, Austria), and their thickness was measured according to ISO 534 with a micrometer F16502 (Frank-PTI, Austria). Handsheets were prepared at the grammage of 75 g/m^2^ and were prepared using a strip cutter for mechanical testing to determine tensile and burst strength indices (ISO 1924-1 and ISO 2758, respectively). 

### 2.6. Tensile Strength Determination

Tensile strength was determined on a Frank Tensile Tester (Frank-PTI, Laakirchen, Austria) according to the International Standard ISO 1924-1:1992(E). 

### 2.7. SEM Analysis

High resolution images were obtained using a scanning electron microscope (SEM). The samples were placed in an Emitech K550X sputter coater (Emitech Ltd., Ashford, United Kingdom) and plated with gold plasma twice. The prepared samples were placed in the scanning electron microscope VEGA TS 5136MM using a voltage of 15 kV and a resolution of 1000×. Images were taken using Vega TC software (version 2.9.9.21) (Tescan R & D, Brno, Czech Republic).

## 3. Results

### 3.1. Analysis of the Raw Material

The chemical composition of the used material is described in a more detailed manner in our previous publication [15]. Birch chips have a great potential for use in the production of 2-furaldehyde due to their glucose (37.8%) and xylose (21.9%) contents, according to feedstock characterization. While glucose-enriched LC residue can also be used as a raw material in pulping procedures to create cellulose-enriched pulp, xylose is a raw material for obtaining 2-furaldehyde. According to the results, 15.4% of the oven-dried mass can theoretically contain the maximum quantity of 2-furaldehyde that can be recovered from the BWC (o.d.m.).

### 3.2. Chemical Composition of Hydrolysis Condensate 

A total of 8 experiments were performed to obtain LC residue for the pulping process. It can be seen in Table 3 that the highest amount of 2-furaldehyde was obtained in experiment No. 1, where the temperature was 175 ℃, the time was 90 min, the catalyst amount was 3 wt.%, and catalyst concentration was 50%. However, the highest amount of LC residue was obtained in experiment No. 2, where temperature was 175 ℃, the time was 60 min, the catalyst amount was 3 wt.%, and the catalyst concentration was 50%.

These results correspond nicely with those from our previous work. We used DesignExpert11 software to optimize this process condition and to obtain LC residue that is highly suited for the pulping process. The increase in temperature and treatment time in obtaining 2-furaldehyde shows a general lowering trend regarding the amount of LC residue left over from the BWC pre-treatment. It must be noted that at higher temperatures, the quality of obtained LC residue is better and more suitable for the pulping process due to the lower amount of admixtures, such as pentosans and low molecular weight extractives [15].

### 3.3. LC Residue Pulping Process

To characterize the LC residues obtained after 2-furaldehyde production, two different pulping methods were used (TMP and APMP pulping processes). The furfural production process conditions were selected as optimal, as determined previously. In this case, the furfural production was utilized as a pre-treatment, with a focus on obtaining the residue. The pre-treatment experiments were split into two groups for two different pulping approaches. The LC residue obtained from the BWC pre-treatment from experiments 1 to 4 (Table 1) was used as a feedstock in the TMP pulping process, while the LC residue from experiments 5 to 8 (Table 1) were used as a feedstock in the APMP process. The obtained pulps for both processes were characterized in terms of carbohydrate and lignin content (Figure 4 and Figure 5).

The obtained data shows that for all pre-treated samples, the TMP pulping process, in combination with pre-treatment, removes most of the xylose found in the sample, while leaving a considerable amount of lignin behind, thus decreasing the quality and usability of these pulps. The lignin content in these samples was between 36 and 37%. This shows that a harsher pulping process is necessary for more complete lignin removal.

In comparison, the APMP pulping process more thoroughly removes lignin from the material, while also removing most of xylose from the sample. This is due to the alkaline/ oxidative conditions in which lignin is deprotonated and much more soluble in the media used, significantly increasing the efficiency of the pulping process. When comparing these two approaches, it can be concluded that while the APMP process is harsher, it is more suited for obtaining LC residue pulping material suitable for paper production. To obtain information about the fiber size and uniformity of the obtained pulps, SEM images were taken. The obtained images were compared to native birch wood (Figure 6). 

SEM images shows that when combining pre-treatment with the TMP pulping process, the obtained fibers are relatively short, with highly varied lengths, while fibers from APMP pulping are longer and more uniform in size. Using SEM images, we determined that for sample 1 TMP, the fiber length was from 30 to 300 µm and for sample 5 APMP, the fiber length was from 400 to 1000 µm. This indicates that for the production of paper grade pulp from LC residue, APMP is the more suitable approach. 

The results of the tensile index from the TMP pulping process can be seen in Figure 7. The highest tensile index (8.5 N·m/g) was for pulp handsheets acquired from LC residue obtained in experiment 4 (catalyst conc. 50%, temperature, 180 °C, and treatment time, 40 min). This result is 5 times higher than the tensile index of the untreated BWC TMP pulp handsheets (1.6 N·m/g). Still, these pulp handsheets are not comparable to standard printing paper, which has a tensile index of 26.3 N·m/g; therefore, they must be used for some other applications.

The results of the tensile index from the APMP pulping process can be seen in Figure 8. The pulp handsheet samples obtained from the APMP process exhibited superior characteristics when compared to the pulp handsheet samples from the TMP process. The best results were obtained for pulp produced from the LC residue obtained in experiment 5 (catalyst conc. 70%, temperature, 175 °C, and treatment time, 60 min). These results can be explained by the higher glucose (cellulose) content in the obtained pulp, as well as the higher fiber length. The tensile index of this pulp was 23.0 N·m/g, which is comparable to standard printing paper. If the LC residue were subjected to deeper delignification, these results could be even higher. It must also be noted that cellulose content on it own does not indicate the tensile index properties of the produced handsheets. The major factor here would be the length of obtained fibers and the degree of degradation of these fibers.

It is very interesting to observe that better tensile strength values are obtained for the LC residues after the pre-treatment process at a moderate (175 °C) temperature when the amount of 2-furaldehyde is 7.85% o.d.m., which is 51% less than the theoretically possible 2-furaldehyde amount (Table 3). The amount of acetic acid in the condensate reached 4.45% o.d.m.

## 4. Conclusions

The obtained results show that it is possible to combine the 2-furaldehyde production from BWC with the pulping process, creating a biorefinery concept. By comparing two pulping methods, it can be seen that the best fiberization results are obtained using the APMP process. The length of the obtained fibers (30–300 µm) using the TMP method are suitable for use in specialized paper production with specific purposes. The length of the obtained fibers (400–1000 µm) using the APMP method can be used for wrapping paper production. Thus, three high-value products can be obtained in noteworthy amounts and quality: 2-furaldehyde (7.85% o.d.m.), acetic acid (4.45% o.d.m.), and pulp with a high tensile index (23.0 N·m/g), comparable to the tensile index of standard printing paper.

## Figures and Tables

**Figure 1 materials-15-05872-f001:**
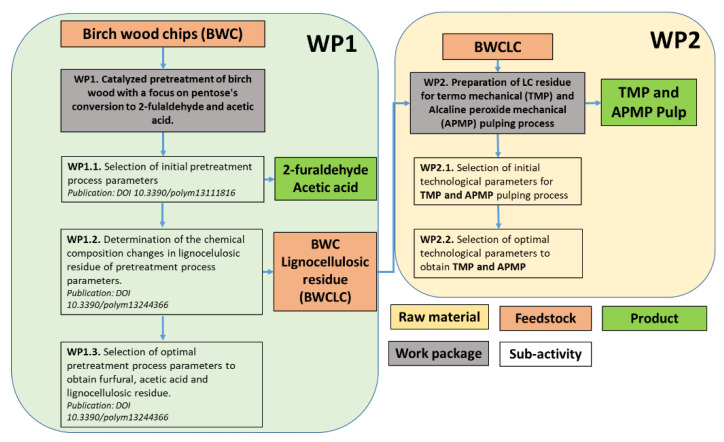
Schematic diagram of the analyzed process.

**Figure 2 materials-15-05872-f002:**
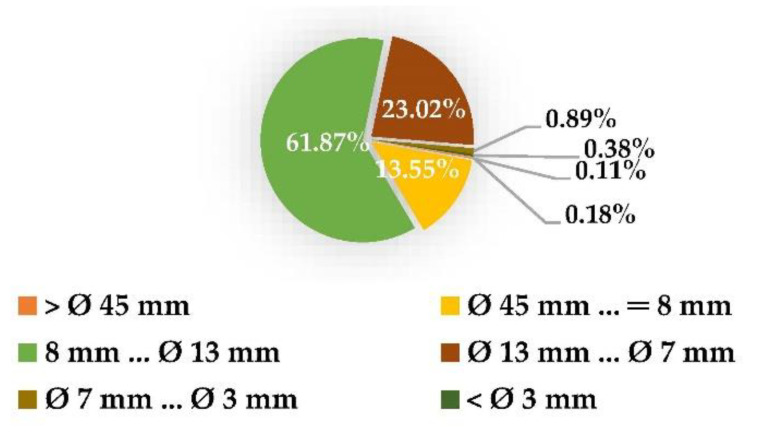
Birch wood chip fractional distribution.

**Figure 3 materials-15-05872-f003:**
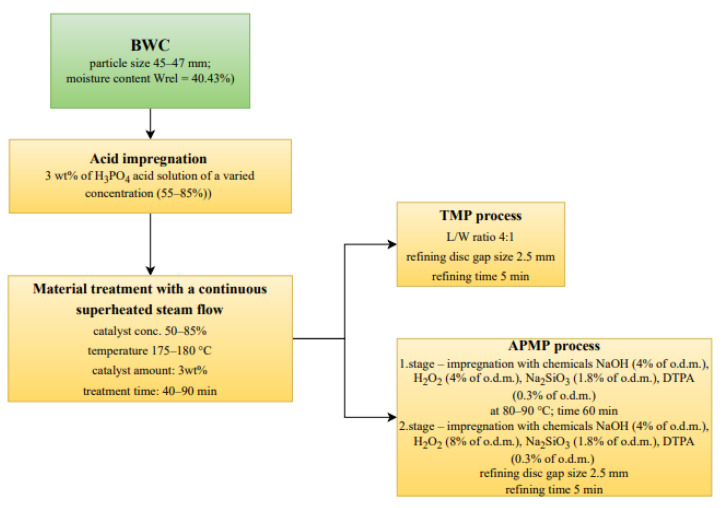
Scheme of the processes of catalyzed pre-treatment of BWC and the pulping process.

**Figure 4 materials-15-05872-f004:**
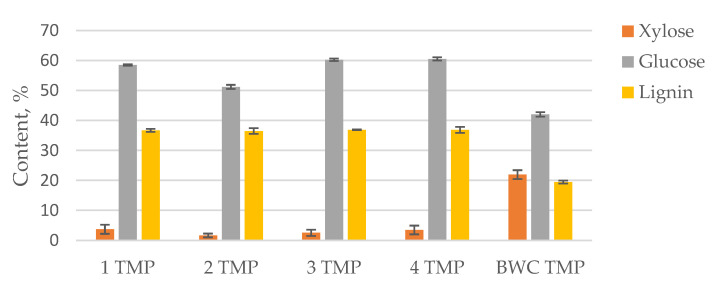
Xylose, glucose, and lignin content in TMP pulps.

**Figure 5 materials-15-05872-f005:**
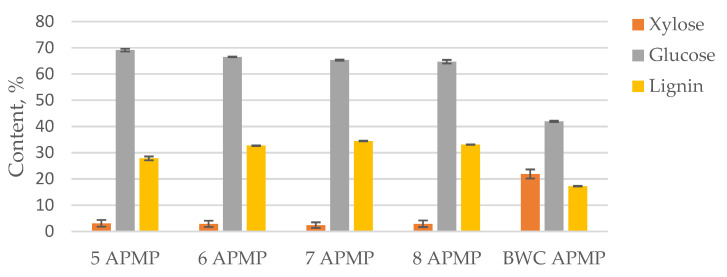
Xylose, glucose, and lignin content in APMP pulps.

**Figure 6 materials-15-05872-f006:**
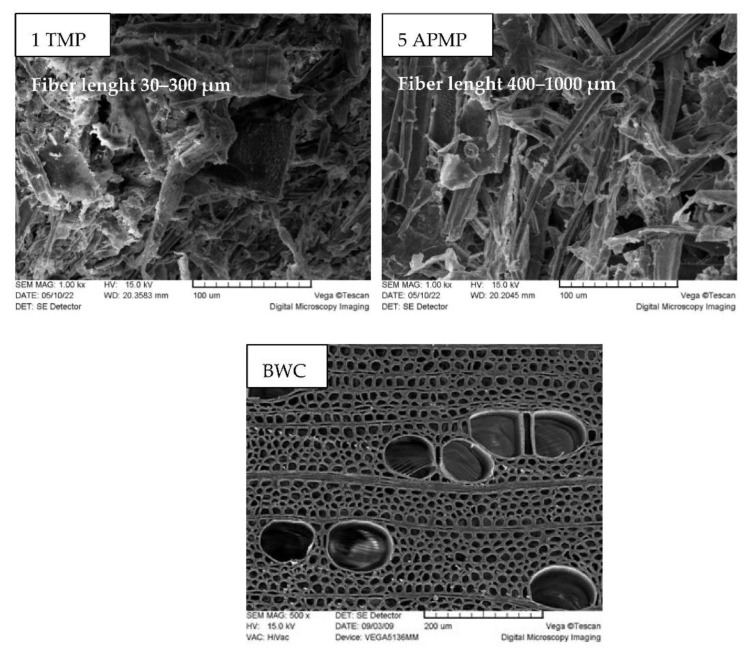
SEM images of untreated BWC, TMP and APMP pulping after BWC pre-treatment.

**Figure 7 materials-15-05872-f007:**
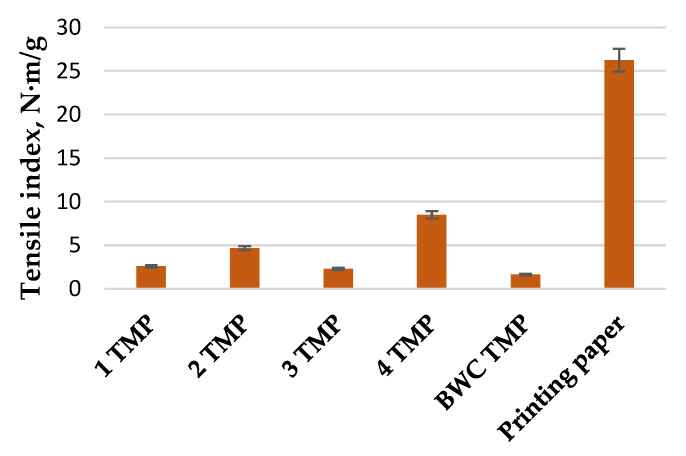
Tensile index of printing paper and pulp handsheets obtained from experiments 1 to 4 of the LC residue TMP pulping process.

**Figure 8 materials-15-05872-f008:**
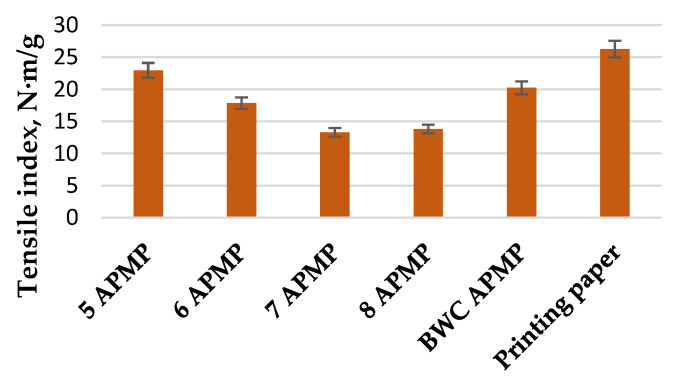
Tensile index of printing paper and pulp handsheets obtained from experiments 5 to 8 of the LC residue APMP pulping process.

**Table 1 materials-15-05872-t001:** Pre-treatment process parameters.

No.	Catalyst Conc. (c)	Temperature (T)	Catalyst Amount (m)	Treatment Time (τ)
	%	°C	wt.%	min
1	50	175	3	90
2	50	175	3	60
3	50	180	3	50
4	50	180	3	40
5	70	175	3	60
6	70	175	3	50
7	85	175	3	60
8	85	175	3	40

**Table 2 materials-15-05872-t002:** Quantities of chemicals using APMP technology.

Stage	Chemicals, % of o.d.m.			
	NaOH	H_2_O_2_	Na_2_SiO_3_	DTPA
1	4.0	4.0	1.8	0.3
2	4.0	8.0	1.8	0.3
Total	8.0	12.0	3.6	0.6

**Table 3 materials-15-05872-t003:** The yield of LC residue and chemical composition of condensate after hydrolysis.

No	Amount, % o.d.m.					
	LC Residue	Formic Acid	Acetic Acid	Levulinic Acid	5-HMF	2-Furaldehyde
1	48.94 ± 0.45	0.19 ± 0.01	2.52 ± 0.01	0.03 ± 0.04	0.03 ± 0.01	10.31 ± 0.10
2	49.14 ± 0.32	0.21 ± 0.02	2.12 ± 0.03	0.02 ± 0.01	<0.01	8.15 ± 0.04
3	46.70 ± 0.36	0.17 ± 0.02	2.23 ± 0.01	0.02 ± 0.01	<0.01	8.32 ± 0.11
4	45.88 ± 0.45	0.02 ± 0.01	1.02 ± 0.01	0.02 ± 0.01	0.02 ± 0.01	7.78 ± 0.04
5	41.72 ± 0.29	0.44 ± 0.11	4.45 ± 0.03	<0.01	<0.01	7.85 ± 0.19
6	41.01 ± 0.34	0.40 ± 0.05	3.81 ± 0.04	<0.01	<0.01	6.40 ± 0.09
7	42.20 ± 0.47	0.45 ± 0.03	4.48 ± 0.11	0.03 ± 0.01	<0.01	7.95 ± 0.07
8	42.65 ± 0.26	0.38 ± 0.01	3.98 ± 0.01	0.02 ± 0.01	<0.01	6.68 ± 0.02

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
