# Peer review of "Thermomechanical and Alkaline Peroxide Mechanical Pulping of Lignocellulose Residue Obtained from the 2-Furaldehyde Production Process"

_materials, 2022, doi:10.3390/ma15175872_

Round 1

Reviewer 1 Report

Dear Authors,

The following are my comments and suggestion to better improve your manuscript.

1. I believe the long-term objective of this study is to remove LC residue from 2-furaldehdye production, as a potential feed for further biorefining process. However, the authors forgot to emphasize what the current manuscript has done and how it will be important for the next steps.

2. There is a disconnect on the justification to develop sustainable LC processing technologies in terms of the demand for the reduction of greenhouse gases. The authors should further articulate this point in the Introduction, first paragraph.

3. The authors should change the title to better give justice to what really is the ain of the study. 

4. To give readers a better perspective on what you are doing, I suggest that you give a schematic diagram to illustrate the bigger picture of the study and emphasize what this specific manuscript will do and how it will help the long-term objective.

5. In the last paragraph of the Introduction, what is the basis of the goal to "obtain more than  60% 2-furaldehyde and more than 10% cellulose"? Is this a standard one or customized to the existing plant?

6. Provide a diagram of the processes of catalyzed pre-treatment of BWC and the pulping process, including the parameters and their values. In this way, the readers are able to easily understand the processes involved, in a more logical manner.

7. The pretreatment process parameters are limited to two variables only, i.e., catalyst concentration and treatment time. The authors should be able to draw conclusion from previous studies or literature on how the other parameters maybe insignificant compared to treatment time and catalyst concentration.

8. The pre-determined 8 experimental design was not discussed how it was derived. Is this based on the Design Expert software? If yes, what was the model used in determining the number of experiments? What are the basis of the model used in Design Expert?

9. Why not consider more temperature and time variables in the hydrolysate condensation? 

10. The APMP process is promising in removing lignin. The authors should give insights as to why this happens? What are the chemical interactions involved? 

11. APMP pulping process gives a higher tensile strength compared to TMP. Authors should give further insights into this phenomenon, by creating or demonstrating the process of APMP in the pulping process vs its tensile strength. 

Author Response

Dear Reviewer,

Best regards,

Maris

Reviewer 2 Report

the article presents method for additional valorisation of biomass and furalaldehyde production.

the paper is well-written and the main message is also clear.

some minor suggestions:

Figure 4. SEM images - the lenght of the fibers can be shown on the images - to better guide the reader on what to look on SEM image. 

Author Response

Dear Reviewer,

Best regards,

Maris

Reviewer 3 Report

The paper is novelty as to obtaining 2-Furaldehyde from lignocellulosic biomass using pretreatment with phosphoric acid. Below are my questions:

Abstract
L12-13 is not clear. Was biomass obtained from 2-furaldehyde?. L14-15 shows the opposite.
L22. What does o.d.m means? I think that oven-dried mass after reading the text. Please insert it in the abstract
Introduction
-  The introduction is very long. L35-57 can be shortened. The topic dealing with climatic change is extensively discussed in the literature.
- L61: "Given this" can be replaced by "Because of that"
- L87-89 should be better written. Pretreatment processes attempt to optimize the separation of lignocellulosic biomass (cellulose, hemicellulose and lignin) to obtain bleached-cellulose and thus increase the yield of second-generation bioethanol.
- Lignin is an inhibitor of enzymatic hydrolysis of cellulose.
- L101-102 is not clearly written. What does means the expression "after 2-furaldehyde"?
- L132-133 Fractional distribution of what in Figure 1. Please, the authors should be more clear to the readers. What does Ø means?
Results
- L222-223: I am not agree with the phrase: The increase in temperature and treatment time in 2-furaldehyde obtaining leads to lower amount of LC residue leftover from the BWC pretreatment.
For example, the increases in treatment time from 50 min (exp. No 6) up to 60 min (exp. No  5) at 175 °C lead to somewhat higher amount of LC residue (41.01 - 41.72). As for temperatures, I can not consider the experiences 1, 2, 3 and 4 because of the pretreatment times differ. That is, I can not take a reference time
between them for comparison. On the other hand, taking into account the exp. No 8 (175°) and exp. No 4 (180°), which have the same time (40 min) but different temperatures (only differing 5°), I also see an increasing in LC residue (42.65-45.88) amount as temperature grows. Only, that enunciated by the authors in L222-223 is proved by exp. No 1 and 2, as to pretreatment time .
Nevertheless, there is certain trend of increasing  2-Furaldehyde's yield with pretreatment time.
- The design of experiment is very poor (Table 1). It does not cover a wide range of temperature and pretreatment time (see DOI: 10.1016/j.indcrop.2013.07.055)
- L225: What kind of admixtures are the authors referring to?
- Why does the xylose content is higher for 1 TMP than for 2 TMP in Fig. 2? Once the yield of 2-Furaldehyde is higher for 1 TMP, it should be expected a lower content of xylose for 1 TMP.
- I am agree with sentence in L244
- The cellulose content does not explain the tensile index of LC residue. I do not see correlation between values obtained in Fig. 2,3 and 5,6. For example is reported that LC in exp. 4 provide the higher tensile index, why should it be?. Considering the values of glucose in Figure 2 both 3 TMP and 4 TMP should give similar values of tensile index in Figure 5.
- What would be the mechanism In this acid pretreatment process to obtain 2-furaldehyde? It is possible a mechanism as reported in DOI 10.1007/s13399-017-0243-0. This literature must be duly cited in the introduction.
- The authors do not report the methodology followed to measure tensile index. It is important to reproduce experiments.
- Why did the authors not include determination of arabinose?  It is the second most abundant pentose in lignocellulosic biomass producing 2-furaldehyde

Author Response

Dear Reviewer,

Best regards,

Maris

Round 2

Reviewer 1 Report

The revised form of the manuscript can be recommended for publication. 

Reviewer 3 Report

The manuscript has been accepted for publication